# The clinical performance of ultra-low-dose shoulder CT scans: The assessment on image and physical 3D printing models

**Ming Lei, Meng Zhang, Niyuan Luo, Jingzhi Ye, Fenghuan Lin, Yanxia Chen, Jun Chen, Mengqiang Xiao** *

Department of Imaging, Zhuhai Hospital Guangdong Provincial Hospital of Traditional Chinese Medicine, Zhuhai, China

* xmqzhuhai@163.com

**Data Availability Statement:** The datasets generated and/or analyzed during the current study are not publicly available due to limitations of

## Abstract

### Objectives

Evaluation of the clinical performance of ultra-low-dose computed tomography (CT) images of the shoulder joint on image-based diagnosis and three-dimensional (3D) printing surgical planning.

### Materials and methods

A total of 93 patients with displaced shoulder fractures were randomly divided into standard-dose, low-dose, and ultra-low-dose groups. Three-dimensional printing models of all patients' shoulder joints were fabricated. The subjective image quality and 3D-printing model were evaluated by two senior orthopedic surgeons who were blinded to any scanning setting. A 3-point scale system was used to quantitatively assess the image quality and 3D printing model, where more than 2 points meant adequate level for clinical application.

### Results

Compared with the standard dose protocol, ultra-low-dose technique reduced the radiation dose by 99.29% without loss of key image quality of fracture pattern. Regarding the subjective image quality, the assessment scores for groups of standard, low, and ultra-low doses were 3.00, 2.76, 2.00 points on scapula and humerus, and 3.00, 2.73, 2.44 points on clavicle. Scores of the three groups for the assessment of 3D printing models were 3.00, 2.80, 1.34 on scapula and humerus, and 3.00, 2.90, 2.06 on clavicle. In the ultra-low-dose group, 24 out of 33 (72.7%) 3D printing models of scapula and humerus received lower than 2 points of the evaluation score, while nearly 94% of the clavicle models reached the adequate level.

### Conclusion

An ultra-low-dose protocol is adequate for the diagnosis of either displaced or non-displaced fractures of the shoulder joint even though minor flaws of images are present. Three-dimensional printing models of shoulder joints created from ultra-low-dose CT scans can be used

ethical approval involving the sensitive patient information and anonymity, but are available from the Corresponding Author, or from ethics committee representative Xiaoyan Li (llbgs@gzucm.edu.cn), on reasonable request.

**Funding:** The author(s) received no specific funding for this work.

**Competing interests:** The authors have declared that no competing interests exist.

for surgical planning at specific bone like the clavicle but perform insufficiently in the overall surgical planning for shoulder injuries due to the significant geometric flaws.

## Introduction

Shoulder injury is primarily examined by conventional radiography and is likely to be further examined by computed tomography (CT) to detect the fracture profile [1]. Computed tomography is superior to radiography for characterizing fracture patterns [2], particularly for regions with anatomic complexity such as shoulder joints. The dose of radiation for traditional CT on a shoulder joint is extremely high with 5.28 mSv in comparison to conventional radiography, which is less than 0.011 mSv [3, 4]. The risk of increased ionizing radiation exposure is the primary safety concern for having a CT scan [5]. According to the data collected in the United States from 1991 to 1996, malignant tumors caused by CT radiation account for 0.4% of all malignant tumors [6–8]. The high dose of radiation of conventional CT scans for patients with bone injuries restricts its application for regular diagnosis. Employing low-dose CT on diagnosis and surgical procedure has been of growing interest in clinical research with the technical advantage of CT scans being to detect detailed anatomical structure. Many researchers targeted the technical solution to reduce the negative effect of low image accuracy acquired from low-dose CT scans [9–12]. Convolutional neural networks and iterative reconstruction algorithms were reported to use for improving image quality [13]. Some published studies confirmed the significantly reduced rates of effective radiation dose, from 49% to 97%, without losing the critical image quality acquired from low-dose CT scans and iterative reconstruction technique [9–12].

A digital three-dimensional (3D) model created from CT scans is not only employed on diagnosis and surgical assistance of bone injury but is also used to create physical models of bone for surgical planning. Bone and joint fractures and anatomical structures of the shoulder can be represented in a 3D-printed model which can be used to develop favorable surgical plans and optimize a surgical protocol. Surgeons have previously employed 3D-printed models to extract the fracture line of a shoulder joint and make a surgical plan for internal fixation implants [14, 15]. Clinicians have utilized 3D-printed models to simulate the surgical procedure of a shoulder joint to guide the implant location and size [16]. Furthermore, a 3D-printed model is an excellent tool for communication between clinicians and patients who can understand the treatment procedure and outcomes even though they lack medical knowledge [17].

Radiation dose of CT scans is positively correlated with CT image quality. There are few studies on the comparative assessment of diagnosis performance among different radiation doses of CT scans. Xiao et al. reported that 3D printed models created from low-dose CT images effectively evaluate the clinical performance of diagnosis and surgical planning [9]. However, assessment of clinical performance on the diagnosis and surgical planning by using 3D printing models created from low-dose shoulder CT images has not been studied. Here, we hypothesize that low dose CT scans meet the clinical needs for fracture diagnosis and 3D printing models. The aim of this study is to evaluate the clinical performance of ultra-low-dose CT images of shoulder joints on the image-based diagnosis and physical model-based (3D printing) surgical planning.

## Materials and methods

### Patient selection

This prospective study was approved by the institutional ethics committee of the (BF2019-030-01). All patients visiting our hospital for CT imaging for the diagnosis of shoulder trauma were

entered into this study. Inclusion criteria were set as follows: age over 18 years, signed written consent for participation, and admission from the emergency department with symptoms of shoulder fracture, non-osteoporosis (bone density: lumbar spine and/or hip joint T-score > -2.5). Exclusion criteria were as follows: less than 18 years of age, pregnancy, patients who refused to participate in the study, and patients with osteoporosis (bone density: lumbar spine and/or hip joint T-score < -2.5) or pathological fractures. Poor image quality of bone would be present on an osteoporosis patient who has significant low bone mass. This may induce poor grey contrast level between bone and surrounding soft tissues.

Patients with shoulder fractures were randomly divided into three groups: a standard-dose group (150 mAs, 120 kV, limb joints No-smart-milliampere) [18], a low-dose group with (105 mAs, 100 kV), and an ultra-low-dose group (52 mAs, 80 kV). The above scanning configurations were set based on the CT Whole Body Phantom "PBU-60 (Kyoto Kagaku Co., Ltd) and pre-experimental CT scans. The CT parameters for each group are listed in Table 1. Written informed consent was obtained from eligible participants.

## Sample size determination

The sample size was calculated using the G*Power software for a priori analysis, with a sensitivity of 95% ($\alpha = 0.05$) and a study power of 90% ($\beta = 0.10$). Since the patients were classified into three groups based on the dose, the effect size was determined by a one-way analysis of variance (ANOVA) using the effect size measure Cohen's f. The resulting sample size n is 12. If 10% missing values were defined, the sample size (n = 15 per group) is sufficient for the study.

## Image acquisition

All selected subjects were scanned using a Toshiba 320-slice dynamic CT scanner (Canon Aquilion One, Japan) with a 0.5 mm slice thickness. The scanning parameters (tube voltage

**Table 1. CT parameters and patient information of each group.**

| Dose group | Standard | Low-dose | Ultra-low-dose | P |
|---|---|---|---|---|
| Tube voltage (kV) | 120 | 100 | 80 | |
| Tube current (mA) | 150 | 140 | 70 | |
| D-FOV (mm) | 500 | 500 | 500 | |
| Rotation time (s) | 1 | 0.75 | 0.75 | |
| Thickness (mm) | 0.5 | 0.5 | 0.5 | |
| Interval (mm) | 0.5 | 0.5 | 0.5 | |
| Scan length (cm) | 160 | 160 | 160 | |
| AIRD3D | Standard | Standard | Standard | |
| No. | 30 | 30 | 33 | |
| Age(year) | 57.17 ± 13.84 | 58.33 ± 19.22 | 59.09 ± 15.96 | 0.90 |
| Gender (Male/Female) | 15/15 | 10/20 | 13/20 | 0.26 |
| Normal bone mass/osteopenia (case) | 14/16 | 11/19 | 13/20 | 0.75 |
| Displaced/Non-displaced fracture (case) | 30/0 | 30/0 | 33/0 | 0.99 |
| Comminuted/simple fracture(case) | 29/1 | 29/1 | 27/6 | 0.99 |
| Clavicle fracture (case) | 12 | 8 | 9 | 0.46 |
| Humerus fracture (case) | 15 | 20 | 20 | 0.26 |
| Scapula fracture (case) | 1 | 1 | 2 | 0.83 |
| Two bones fracture (case) | 2 | 1 | 2 | 0.91 |

D-FOV: Display Field of View

AIRD3D: Adaptive Iterative Dose Reduction

and current) for each group are shown in Table 1. Patients' images were reconstructed by using adaptive iterative dose reduction (3D standard; Canon Medical Systems). Two dose metrics, volumetric CT dose index (CTDIvol; mGy) and dose length product (DLP; mGy*cm) were reported. The effective dose (ED; mSv) for each examination was calculated by multiplication of the DLP and a body region-specific conversion factor k (mSv/DLP) for the hip [19].

## Preparation of three-dimensional models

Patients' 3D models of shoulder joints were reconstructed from the original CT data stored in DICOM format by using Mimics Research 19.0 (Materialise, Belgium). The threshold value was set to "soft tissue (CT) 26-Max" for bone reconstruction. These 3D models (saved as MCS format) were further transferred in Gcode format by using specific software provided by the manufacturer of the 3D printer (Tianwei ColiDo 3.0, China). It took approximately 15.5 hours to print each model. The printing material was polylactic acid. The resolution of the printed models was 0.011 mm × 0.011 mm × 0.0025 mm.

## Types of fractures in each group

In this study, shoulder fractures were divided into simple fractures and comminuted fractures (defined as fractures in which the bone was broken with 3 or more pieces) in terms of the strategy of treatment. Non-surgical treatment is mostly used for simple fracture, which is also treated by surgery in some relatively severe cases. Comminuted displaced fractures have to be treated by surgery [20]. According to the severity of fracture displacement, fractures were also split into displaced and non-displaced fractures. Non-displaced fractures are defined as having no angulation or shortening, a fracture line less than 2 mm wide and/or less than 1 mm displacement of the bone cortex. Displaced fractures are defined as having a fracture line more than 2 mm wide and/or more than 1 mm displacement of the bone cortex. Avulsion fractures caused by a sudden and violent contraction of a muscle or ligament, were grouped into non-displaced fractures or displaced fractures when the fracture had a long-diameter ≤ 5 mm or > 5 mm wide bone piece, respectively [21].

## Assessment of image quality

Two senior radiologists, who were part of a national board of radiologists and were clinicians with over 10 years of clinical experience for clinical diagnosis of musculoskeletal diseases performed quality assessment of CT images. They were blinded to the results of the study.

The objective CT image quality metrics were performed by a senior radiologist. An oval shape of region of interest (ROI) was placed on the thickest region of the cross section of the cortical shell of the shoulder bones. The oval ROI was equal to 15 $mm^2$. A circular ROI with 100 $mm^2$ was placed within muscle. Computed tomography values of shoulder muscle (CTm), and shoulder cortical bone (CTc) were determined. The contrast-to-noise ratio (CNR) and signal-to-noise ratio (SNR) were calculated in terms of following equations [9]:

Signal-to-noise ratio = CTm mean / standard deviation (SD) mean
Contrast-to-noise ratio = (CTc mean–CTm mean) / SD mean.

The assessment of the subjective image quality of the fracture profile, which we hereafter termed as the "fracture line score", was performed by two radiologists with rich experience in diagnosing musculoskeletal diseases. They were blinded to the scanning parameters and rated the image quality based on a 3-point score system as previously reported [9]: 3 = good (good visualization of fracture line, excellent definition of fracture profile, see Figs 1 and 2); 2 = adequate (adequate visualization of fracture line, slightly impacted by the image noise, good

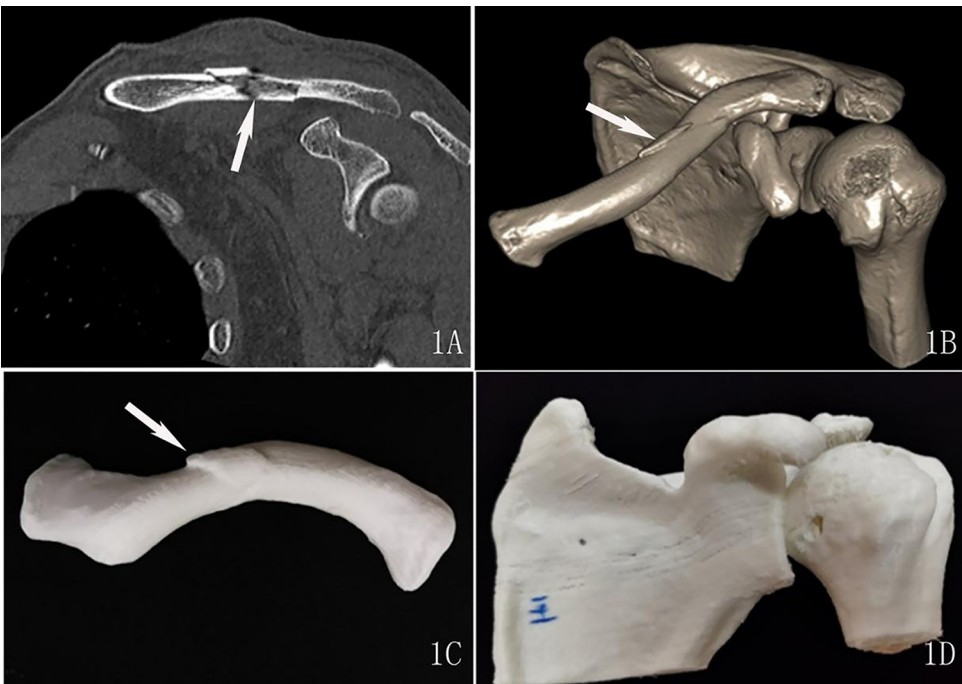

**Fig 1. A 51-year-old man with a left clavicle fracture underwent a standard-dose CT scan.** 1A: oblique coronal reconstruction images show the fracture line (arrow), and scored 3 points. 1B: Three-dimension reconstruction images show the fracture lines (arrow), 3 points. 1C and 1D: Three-dimension printed model, scored 3 points.

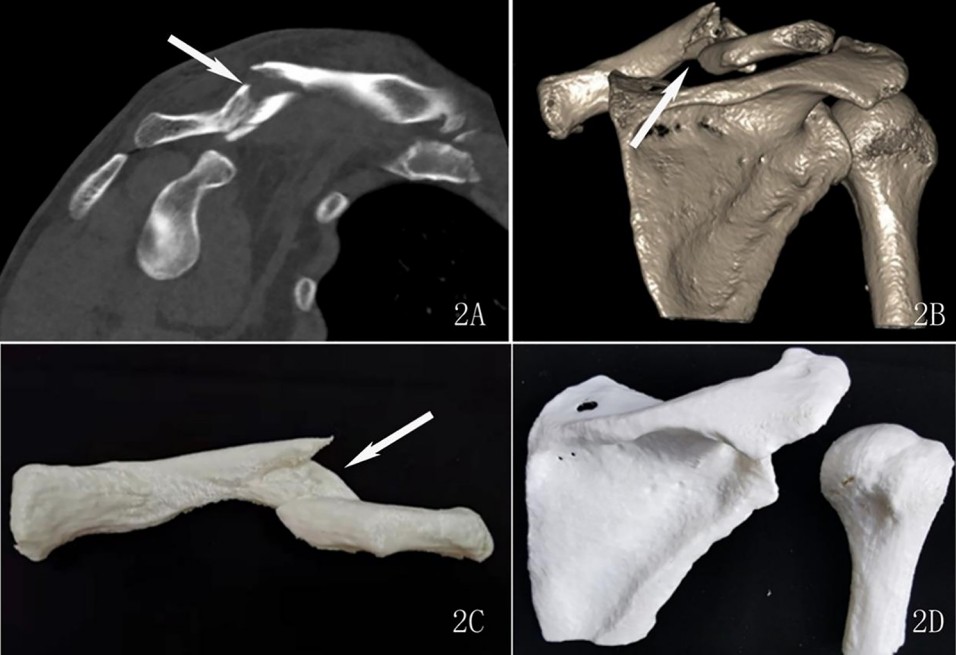

**Fig 2. A 49-year-old man with a left clavicle fracture underwent a low-dose CT scan.** 2A: oblique coronal reconstruction images show the fracture line (arrow), scored 3 points. 2B: Three-dimension reconstruction images show the fracture lines (arrow), scored 3 points. 2C and 2D: Three-dimension printed models of the clavicle and the scapula and humerus, respectively, scored 3 points.

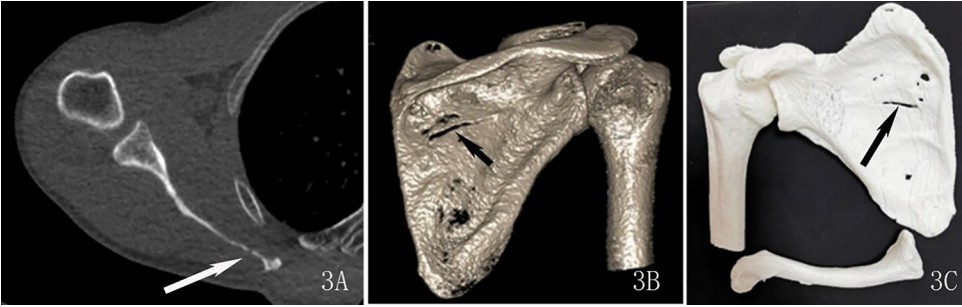

**Fig 3. A 58-year-old man with a scapula body linear fracture underwent an ultra-low-dose CT scan.** 3A: axial images show the fracture line (arrow), scored 2 points. 3B: Three-dimension reconstruction images show the fracture line (arrow), scored 2 points. 3C: Three-dimension printed model, scored 2 points.

definition of fracture profile, see Fig 3); 1 = poor (inadequate visualization of fracture line, poor definition of fracture profile, see Fig 4).

## Assessment of three-dimensional printing models quality

The assessment of 3D printing models was performed by two senior orthopedic surgeons who were blinded to the scanning parameters. The evaluation criteria were based on accuracy and clarity of anatomic structure for surgical planning purposes. A 3-point rating system developed by published studies was employed for the assessment: 3 = good (smooth surface of a model

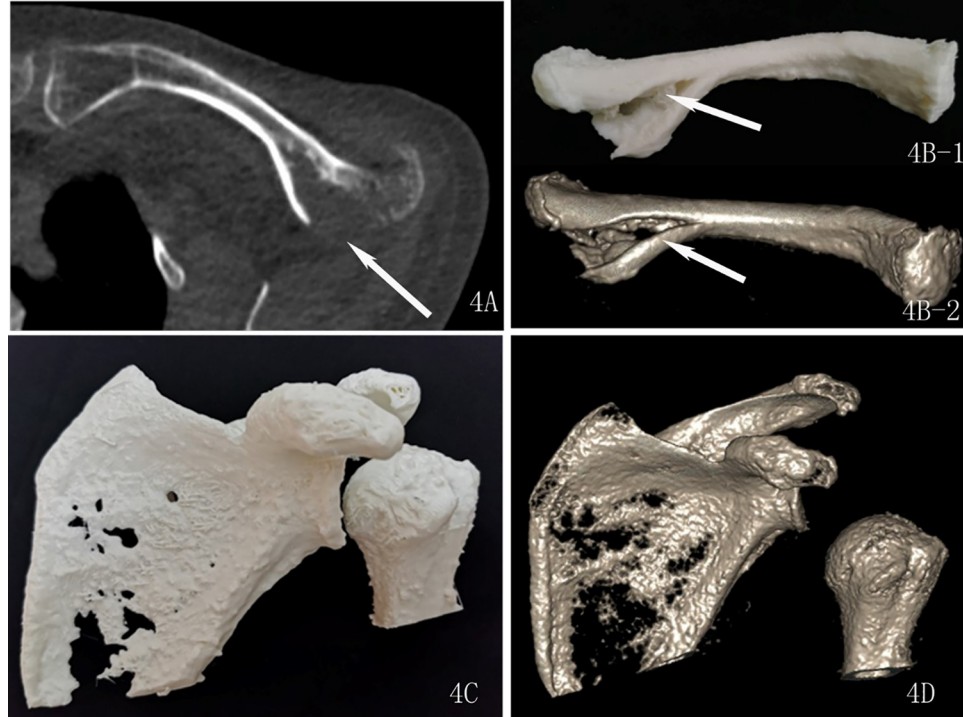

**Fig 4. A 65-year-old female with a left clavicle comminuted fracture underwent an ultra-low-dose CT scan.** 4A: axial image shows the fracture line (arrow), scored 2 points. 4B and 4D: Three-dimension reconstruction images of the clavicle show the fracture line (arrow). 4B: Three-dimension reconstruction images, scored 2 points. 4C: Three-dimension printed model of the scapula and humerus, scored 1 point.

with high similarity of anatomic structure, quite applicable for surgical planning); 2 = adequate (slightly coarse surface of a model with minor geometric flaws, applicable to the basic surgical planning not affected by the minor flaws); 1 = poor (coarse surface of a model with distinct geometric flaws, not applicable to surgical planning) [9, 22, 23].

## Gold standard of diagnosis performance

The gold standard of diagnosis performance is surgical findings or computed tomography/ Diagnostic Radiology re-examination. For fractures treated with surgery, the diagnosis was made by observing the fracture directly during surgery. For fractures by expectant treatment, the diagnosis was made based on the CT/DR review within 1–3 months: callus at the fracture end; dysplasia or old fracture without callus.

## Statistical analysis

The statistical package SPSS version 26.0 software (IBM Corp, Armonk, NY, USA) was used for all statistical analyses. Continuous data were expressed as means±SD. Comparisons among the standard-dose, low-dose, and ultra-low-dose groups were conducted by using independent sample ANOVA tests. The Tamhane's T2 test was employed to assess the difference in quality scores of CT images and 3D-printed models. The intraclass correlation coefficient (ICC) test was used to analyze the interobserver agreement in the qualitative evaluation. An ICC value > 0.8 was considered as agreement.

## Results

This study was comprised of 93 patients with a mean age of 58.33 years (range, 18–89 years). A total of 55 (59.1%) of the patients were male. There were no statistical differences in age or gender (Table 1). Patients with normal bone density or osteopenia were randomly assigned in the standard, low-dose, and ultra-low-dose groups and generated the following distribution: 14 (normal bone density)/16 (osteopenia), 11/19, and 13/20 respectively. All patients suffered a displaced fracture (Table 1). The standard and low-dose group had 1 simple and 29 comminuted fractures while the ultra-low-dose group had 6 simple and 27 comminuted fractures (Table 1). Fractures were present in three important bones of the shoulder joint, clavicle, humerus, and scapula, in all groups. The distribution of clavicle fractures of three groups (from standard to ultra-low-dose groups) were 12, 8, 9. Fractures of the humerus and/or scapula in three groups were 18, 22, 24, as shown in Table 1.

**Table 2. Objective image evaluation of the different groups.**

| Dose group | Standard | Low-dose | Ultra-low-dose | P |
|---|---|---|---|---|
| CTDIvol (mGy) | 15.263 ± 0.605 [a] | 6.660 ± 0.257 [b] | 1.654 ± 0.066 [c] | 0.001 |
| DLP (mGy*cm) | 229.733 ± 29.087 [a] | 101.586 ± 12.487 [b] | 25.154 ± 2.884 [c] | 0.001 |
| ED (mSv) | 3.216 ± 0.328 [a] | 1.086 ± 0.135 [b] | 0.228 ± 0.026 [c] | 0.001 |
| CTc | 1,737.30 ±101.01 [a] | 1,897.84 ± 130.42 [b] | 2,124.72 ± 264.15 [c] | 0.001 |
| CTc-CTm | 1,675.79 ± 97.31 [a] | 1,840.07 ± 128.59 [b] | 2,058.25 ± 272.21 [c] | 0.001 |
| SD | 15.11 ± 37 [a] | 22.47 ± 6.17 [b] | 33.53 ± 8.28 [c] | 0.001 |
| SNR | 57.99 ± 12.44 [a] | 39.00 ± 12.27 [b] | 26.14 ± 7.95 [c] | 0.001 |
| CNR | 114.81 ± 22.82 [a] | 87.22 ± 22.90 [b] | 65.16 ± 18.18 [c] | 0.001 |

Data are presented as mean ± standard deviation.

Analysis of variance (F-test) between three groups. P < 0.05 means statistically significant. [a]P = 0.001 vs Low-dose; [b]P = 0.001 vs Ultra-low-dose; [c]P = 0.001 vs Standard.

**Table 3. Subjective evaluation of the different groups.**

| Dose group | Bone | Standard | Low-dose | Ultra-low | P |
|---|---|---|---|---|---|
| Facture line score | Clavicle | 2.999 ± 0.001[a] | 2.7270 ± 0.467[b] | 2.444 ± 0.527[c] | 0.001 |
| | Scapula and Humerus | 2.999 ± 0.001 [a] | 2.761 ± 0.436 [b] | 2.000 ± 0.001 [c] | 0.001 |
| 3D model score | Clavicle | 2.999 ± 0.001[a] | 2.900 ± 0.305[a] | 2.062 ± 0.435[b] | 0.001 |
| | Scapula and Humerus | 2.999± 0.001 [a] | 2.800 ± 0.406 [b] | 1.343 ± 0.482 [c] | 0.001 |

Data are presented as mean ± standard deviation.

Clavicle, Scapula and Humerus facture line score: Analysis of variance (F-test); F = 23.85–1049.35, [a]P = 0.001 vs Low-dose; [b]P = 0.001 vs Ultra-low-dose; [c]P = 0.001 vs Standard. Clavicle 3D model score value: Analysis of variance (F-test); [a]P = 0.078 vs Low-dose, $F = 3.22$; [b]P = 0.001 vs Ultra-low-dose, F = 75.99; [c]P = 0.001 vs Standard, F = 38.98. Scapula and Humerus 3D model score value: [a]P = 0.009 vs Low-dose, F = 7.25; [b]P = 0.001 vs Ultra-low-dose, F = 163.92; [c]P = 0.001 vs Standard, F = 353.04.

The radiation dose of the ultra-low dose protocol was reduced by 99.29% compared with standard counterpart, 0.228 vs 3.216 on effective dose values. The mean volumetric CT dose index of the three groups (from standard to ultra-low-dose) was 15.26, 6.66, and 1.65 respectively, and the mean DLP (mGy*cm) values were 229.73, 101.59, and 25.15 respectively, as shown in Table 2.

The fracture line scores, which represented the subjective image quality, of the three bones of the shoulder (the scapula, humerus, and clavicle) were positively dependent on the radiation dose of images and dropped from 3 to 2 as the radiation dose decreased across the three groups (Table 3). The fracture line score of clavicles with ultra-low-dose was 2.4, while the standard dose was given a higher score of 3.

Three-dimension printed models in the standard dose group scored 3 points (see Fig 1). The score of a 3D printing model declined as the radiation dose reduced. The clavicle models of the ultra-low-dose group had a mean of 2.06 on the evaluation score while the 3D printing models of the scapula and humerus created from ultra-low-dose CT scans received a lower score (1.34) (see Fig 4) (Table 3).

The ICC values for the evaluation scores on 3D printed models and fracture lines of the clavicle were 0.889 and 0.872, respectively. The same ICC value (0.872) was obtained from the evaluation scores on 3D printed models and fracture lines of the scapula and humerus.

## Discussion

This is the first study to perform an assessment on the clinical performance of ultra-low-dose radiation on the image and physical model of shoulder joints. Assessments covered ultra-low-dose to standard dose to investigate the comparative performance among different radiation doses. Three-dimension printing physical models of shoulder joints were firstly used to evaluate the radiation-dose-related clinical performance for shoulder injuries. A combination of image and real entity is a technical method to help provide an answer to patients' and clinicians' concerns regarding an optimal balance between radiation-related health risk, diagnostic accuracy, and surgical optimization for shoulder injuries.

Sound clinical performance was present in the 3D printing models of the clavicle created from ultra-low-dose CT scans. Favorable accuracy of anatomic structure allowed those models to score high on surgical planning, with more than 2 points on the evaluation score given by senior radiologists. Experimental results demonstrated that approximately 94% of these 3D printed models had sufficient accuracy in terms of anatomic detail to perform surgical planning. There are no published studies investigating the clinical performance of ultra-low-dose CT imaging on shoulder treatment strategy.

We have previously demonstrated that the quality of CT images at the wrist joint meet the needs of clinical fracture diagnosis and the image quality of the 3D model created from ultra-low-dose CT scans is good enough for the surgical planning [9]. However, in terms of the scapula part of the shoulder joint, the image quality of ultra-low dose CT meets the needs of clinical fracture diagnosis, but the image quality of the 3D model is too low for the surgical planning. Despite the favorable performance of an ultra-low-dose protocol on the clavicle model, the ultra-low-dose protocol did not extend its clinical benefit on the 3D-printed model of the scapula and humerus. These models were scored much lower (less than 1.5 out of 3) by professional experts, and nearly 73% of the 3D-printing models on these bones were inapplicable to clinical planning, suggesting a failure of clinical application of 3D-printing models of the scapula and humerus by using an ultra-low-dose protocol. To our knowledge, an extremely thin layer of cortical bone of the humeral head and scapula body may be the reason for poor gray level contrast development with respect to the surrounding soft tissues, particularly imaging under ultra-low radiation dose (Fig 4). There is no doubt that a high radiation dose of CT scans leads to the development of higher-definition images.

The conventional orthopaedic surgery system does not have high requirements for 3D printing accuracy, which is enough to reach 0.1 mm. In this subject, extrusion 3D printing (fused deposition modeling, FDM) is the cheapest of all 3D printing, and the 3D printing accuracy is 0.011 mm, which fully meets the demand. Higher precision 3D printing is too high cost and is not conducive to research, making it difficult for 3D printing technology to be popularized in hospitals. The relatively low-cost material PLA (Tianwei Co., Ltd., US $9.83 for 1KG PLA material) was used. Printing one 3D model costs about $2.16 and takes about 15.51±1.20 hours. In our study, 3D printing process self-study videos were provided by 3D printing manufacturers, and 3D modeling parameters were adjusted remotely by the manufacturers. Ordinary computers could run the software, which was easy to learn.

The ultra-low-dose protocol functioned well in the diagnosis of fracture pattern of an injured shoulder. The ultra-low-dose protocol scored more than 2 points in subjective image quality (Table 2), which suggests that a good balance between low health risk and acceptable diagnosis accuracy can be reached. Alagic et al. [24] found that ultra-low-dose CT was a sound alternative to conventional radiography in the diagnosis of peripheral skeleton injury (wrist, ankle, and knee) as it had distinct technical strength in providing a detailed fracture pattern inside the bone as well as a comparable radiation dose. It is not possible to suggest that ultra-low-dose radiation is superior to the conventional standard dose in terms of clinical application as the latter received a full score in the evaluation of subjective image quality.

A significant reduction in the radiation dose, by using an appropriate setting of scanning parameters, can significantly reduce the negative influence of dose-related image quality of the shoulder joint. The ultra-low-dose technique employed in this study reduced the tube voltage by 33% and halved the tube current when compared to the standard-dose protocol. Furthermore, the algorithm of adaptive iterative dose reduction used in this study effectively worked on noise reduction and improvement of image quality [9, 25, 26]. These technical means reduced the effective radiation by 99.29%. Interestingly, the image quality was not visually faded and was still well workable on the diagnosis. Further evaluation of the subjective image quality reached a favorable level, more than 2 out of 3 points of the assessment score. Notably, assessment of image quality of the shoulder joint, either on the definition of the type of fracture or the severity of fracture, matched the gold standard developed from surgical findings. There were precedents reported in published studies with focused on examination of extremities (shoulder, pelvis, ankle, and wrist) where the radiation dose was reduced to 50% [3].

There are some limitations in this research. Firstly, an in-depth assessment of the diagnosis performance of an ultra-low-dose protocol on acute shoulder injuries, where high-definition

images were required, was not conducted in this study. Furthermore, the 3D printing technique would not be recommended for emergency diagnosis or treatment of shoulder injury due to the time consuming printing procedure and expensive fabrication cost. It should be noted that the image quality is associated with a variety of CT scanner hardware and software, such as tube or detector material, iterative reconstruction algorithm, etc. Radiation dose is not the only key component to evaluate the image quality.

## Conclusions

An ultra-low-dose protocol is adequate for the diagnosis of either displaced or non-displaced shoulder joint fractures even though minor flaws in the images are present. Three-dimension printing models of the shoulder joint created from ultra-low-dose CT scans can be used in the surgical planning of specific bones like the clavicle but perform insufficiently in the overall surgical planning for shoulder injuries due to the significant geometric flaws. Radiation dose should increase in order to obtain high-definition CT images for developing accurate 3D printing models.

## Supporting information

**S1 Checklist.**
(DOC)

**S1 File.**
(PDF)

## Author Contributions

**Conceptualization:** Niyuan Luo, Jun Chen.

**Data curation:** Ming Lei, Meng Zhang, Niyuan Luo, Jingzhi Ye, Fenghuan Lin, Yanxia Chen, Mengqiang Xiao.

**Formal analysis:** Ming Lei, Mengqiang Xiao.

**Validation:** Meng Zhang, Niyuan Luo, Jingzhi Ye, Fenghuan Lin, Yanxia Chen.

**Visualization:** Meng Zhang, Niyuan Luo, Jingzhi Ye, Fenghuan Lin, Yanxia Chen.

**Writing – original draft:** Ming Lei, Mengqiang Xiao.

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
