## [Decision Letter · Decision Letter 0]

22 Apr 2022

PONE-D-22-07553The clinical performance of ultra-low-dose shoulder CT scans: the assessment on image and physical 3D printing modelsPLOS ONE

Dear Dr. Xiao,

Thank you for submitting your manuscript to PLOS ONE. After careful consideration, we feel that it has merit but does not fully meet PLOS ONE’s publication criteria as it currently stands. Therefore, we invite you to submit a revised version of the manuscript that addresses the points raised during the review process.

We look forward to receiving your revised manuscript.

Kind regards,

Jonas Bianchi, DDD, MS, Ph.D

Academic Editor

PLOS ONE

Journal Requirements:

Additional Editor Comments:

Thank you for submitting your paper for consideration.

Please, see some points that needs major revision.

1. Please, state the hypothesis of your study (null and/or alternative)

2. Please, add the sample size calculation and study power of your data.

3. ". Assessment of clinical performance on the diagnosis and surgical planning by using 3D printing models created from low-dose CT images has not been reported in published studies."

Are you sure about this information? What are you referring to here? I can find many studies on related topics. Please, be more specific.

4. Was the 0.5 slice thickness, but and the other dimensions? please provide the resolution for each dimension of your voxel.

5. "Patients’ 3D models of shoulder joints were reconstructed from the original CT data stored in DICOM format by using Vitrea fX"...

What is this? You need to use either a reference or named it with the name of the software, company, state, country etc.,

6. What was the resolution of the printed models in x,y and z? this information is essential to your paper.

7. "Two senior radiologists, with over 10 years of clinical experience and who were blinded to the results, performed quality assessment of CT images..."

What do you mean to say by senior? Were those radiologists part of a national board of radiologists? Faculty? Researchers? Clinicians? I need to know how to asses their expertise, rather than "senior"

8. "hey were blinded to the scanning parameters and rated the image quality based on a 3-point score system developed by published studies"

Please, specify which study using references.

9. " A 3-point rating system developed by published studies was employed for the assessment"

Same comment as before. If the study that you're referring to is the number 5, is this study (5) a validation study? It seems to be a validation study because you are using it as gold standard for your scores.

10. Statistical analysis:

Please, in addition to the ICC add a Bland-Altman test for inter-observer agreement for each group.

11. "Three-dimension printing models of the shoulder joint created from ultra-low-dose CT scans can be used in the surgical planning of specific bones like the clavicle but perform insufficiently in the overall surgical planning for shoulder injuries due to the significant

geometric flaws "

I don`t see that yous study has proven the hypothesis that it can be used for surgical planning, since you have not tested this. Please, re-write your conclusions in the paper and abstract based on your results only.

12. "Gold standard of diagnosis performance: The gold standard of diagnosis performance is surgical findings or computed tomography/Diagnostic Radiology re-examination"

Please, provide more information, references, and how the gold standard was performed

13: "Radiation dose must increase to obtain high-definition CT images for developing accurate 3D printing models."

This is a general idea, that should be state with caution, and not as an affirmation.

14: Tables: The table names needs to contain the statistical test that you did. None of them has this information. "P<0.05 means statistically significant" should go in the footnote.

Your ANOVA table contains no letters to distinguish differences between and among groups.

Reviewers' comments:

Reviewer's Responses to Questions

**Comments to the Author**

1. Is the manuscript technically sound, and do the data support the conclusions?

Reviewer #1: Yes

Reviewer #2: Yes

Reviewer #3: Yes

2. Has the statistical analysis been performed appropriately and rigorously? 

Reviewer #1: Yes

Reviewer #2: Yes

Reviewer #3: Yes

3. Have the authors made all data underlying the findings in their manuscript fully available?

Reviewer #1: Yes

Reviewer #2: Yes

Reviewer #3: Yes

4. Is the manuscript presented in an intelligible fashion and written in standard English?

Reviewer #1: Yes

Reviewer #2: Yes

Reviewer #3: Yes

5. Review Comments to the Author

Reviewer #1: Congratulations for the work, I believe that its relevance is justified as it seeks a way to reduce the radiation dose for patients and at the same time obtain a good diagnosis on a day-to-day basis.

- On the first paragraph of the introduction after the sentence “The dose of 50 radiation for traditional CT on a shoulder joint is extremely high with 5.28 mSv in 51 comparison to conventional radiography, which is less than 0.011 mSv...” I suggest adding the information of why you are studying the ultra-low dose CT and why it is so important to try to reduce the radiation dose in patients. Suggested references to justify the importance of studying the ultra-low dose protocol of CT:

o Oestreich AE. RSNA centennial article: ALARA 1912: "As low a dose as possible" a century ago. Radiographics. 2014 Sep-Oct;34(5):1457-60. doi: 10.1148/rg.345130136. PMID: 25208291.

o Mathews JD, Forsythe AV, Brady Z, Butler MW, Goergen SK, Byrnes GB, Giles GG, Wallace AB, Anderson PR, Guiver TA, McGale P, Cain TM, Dowty JG, Bickerstaffe AC, Darby SC. Cancer risk in 680,000 people exposed to computed tomography scans in childhood or adolescence: data linkage study of 11 million Australians. BMJ. 2013 May 21;346:f2360. doi: 10.1136/bmj.f2360. PMID: 23694687; PMCID: PMC3660619.

- The section “Patient selection”: Why did you include patients with osteoporosis in your study? As the bone density is altered in these individuals and their CT image can be different (or worse) from individuals without osteoporosis.

- “Preparation of three-dimensional models”: In this section, information about the 3D printing method was missing, such as: type of 3D printing, reason for choosing this type of printing, diameter of the polylactic acid filament. The printing method was by extrusion (FDM) in your study, as I understand it, but we know that the printing quality using the Digital Lighting Process (DLP) method with vat polymerization is more accurate and faster (although it is more expensive). By adding this information, you can cite Eltes et al.1 study that compared 3D physical models printed with FDM or DLP, and demonstrated that the surface qualities, measured by roughness are adequate (~99% of values <0.1 mm) for both physical models, for their anatomic region. So, this justifies your choice for FDM 3D printing process. But, as in your article the printing was of low quality for some regions, it might be interesting for a next approach to try another type of 3D printing with greater precision (DLP, for example), for those regions that did not perform well with ultra-low dose CT 3D models.

o 1- Eltes, PE, Kiss, L, Bartos, M, et al. Geometrical accuracy evaluation of an affordable 3D printing technology for spine physical models. J Clin Neurosci 2020; 72: 438–446.

- “Types of fractures in each group” section: Although the term comminuted fracture is generally understood, it would be interesting to briefly define the term after mentioning it for the first time.

- Lines 145 and 155: You mention the 3-point score system developed by “published studies”, but you only cite one study. If there are other studies, you must cite more then one (you cited only one – reference 5), and if you want to cite only this reference (5), then you should put this statement in the singular.

- Discussion section:

o Line 223: You mention “Experimental results”, which are they? What is the reference for this affirmation?

o In your study, the ultra-low-dose protocol of the scapula and humerus on the 3D-printed model was inapplicable to clinical planning and failed its clinical application. Xiao et al.2, in 2021, obtained 3D printing models that, despite the quality being lower in the ultra-low dose group, were still sufficient to contribute to the preoperative evaluation and study, the diagnostic performance was not affected by the ultra-low dose protocol.

To what do you attribute this difference between studies? Do you attribute to the different anatomical regions being studied and their particularities? It would be interesting to add information like this to the discussion.

2- Xiao M, Zhang M, Lei M, Hu X, Wang Q, Chen Y, et al. Application of ultra-low dose CT in 3D printing of distal radial fractures. Eur J Radiol. 2021;135:109488. Epub 2021/01/02. doi: 10.1016/j.ejrad.2020.109488. PubMed PMID: 33385624.

o Line 271:

FDM 3D printing, especially of large anatomical regions like the shoulder, take time, however, in terms of cost, it is currently much more affordable and 3D printing by extrusion (FDM) is the cheapest of all 3D printing. The advantage that 3D printing offers for this type of fracture, in terms of diagnosis and treatment accuracy, does it not outweigh the costs of 3D printing? I suggest you address this in the discussion. For example, Guochen Luo et al.3, for their anatomic region of interest, observed the following advantages from 3D printing technology assistance: less trauma, short operation time, less bleeding and reducing the difficulty of operation, which can reduce the waste of bone graft, and more complete reconstruction of the anatomical structure of the defective bone.

3- Luo G, Zhang Y, Wang X, Chen S, Li D, Yu M. Individualized 3D printing-assisted repair and reconstruction of neoplastic bone defects at irregular bone sites: exploration and practice in the treatment of scapular aneurysmal bone cysts. BMC Musculoskelet Disord. 2021 Nov 25;22(1):984. doi: 10.1186/s12891-021-04859-5. PMID: 34

Reviewer #2: This manuscript describes The clinical performance of ultra-low-dose shoulder CT scans: the assessment on image and physical 3D printing models.

I have some comments.

Material and Methods

patient selection

1.line 87. “osteoporosis (bone density: lumbar spine and/or hip joint T> -2.5).” . I think this is the definition of normal bone and osteopenia It could be write non-osteoporosis.

2.line 88, 90 “ T> -2.5” “ T< -2.5” . It could be write as “T-score > -2.5, T-score< -2.5”

3. How about the pathological fracture patient? Were these patients included in the study?

Types of fractures in each group

line 125. I think the definition of “avulsion fracture” means bone was avulsed by the ligament or tendon insertion. This description maybe confused by the reader. Please clarify this description.

line 127. I think the definition of “dislocation” means the relationship of the joint. It could be write as “ displacement of the bone cortex”.

Results

line 197. “ (Table 2)” should be “(Table 3)”, please check it

Table 1

Age (year) “ ±” is redundant

Gender Dose the author mean male/female 15/15 10/20 13/20? Please clarify

Normal bone 14/16, 11/19/13/20, this description was confused by the reader, please clarify

Reviewer #3: - The manuscript raises an interesting topic, but some corrections may make the information clearer.

- In the abstract it is described that the patients were randomly divided into two groups (standard-dose and ultra-low-dose groups), although in the materials and methods three are described. Please correct this information.

- I suggest reviewing Table 01, there are different types of information. Please make it clearer.

6. PLOS authors have the option to publish the peer review history of their article (what does this mean?). If published, this will include your full peer review and any attached files.

Reviewer #1: No

Reviewer #2: No

Reviewer #3: No

---

## [Author Response · Author response to Decision Letter 0]

21 Jul 2022

Dear Dr. Bianchi,

Thank you very much for your decision letter and advice on our manuscript (Manuscript No. PONE-D-22-07553) entitled “The clinical performance of ultra-low-dose shoulder CT scans: the assessment on image and physical 3D printing models”. We also thank the reviewers for the constructive comments and suggestions. We have revised the manuscript accordingly, and all amendments are indicated by red font in the revised manuscript. In addition, our point-by-point responses to the comments are listed below this letter.

This revised manuscript has been edited and proofread by Medjaden Inc..

We hope that our revised manuscript is now acceptable for publication in your journal and look forward to hearing from you soon. 

With best wishes,

Yours sincerely,

Mengqiang Xiao

First of all, we would like to express our sincere gratitude to the reviewers for their constructive and positive comments.

Please, see some points that needs major revision.

1. Please, state the hypothesis of your study (null and/or alternative)

Response: The following sentence was added in the Introduction section.

“We hypothesize that low dose CT scans meet the clinical needs for fracture diagnosis and 3D printing models.”

2. Please, add the sample size calculation and study power of your data.

Response: The following sentences were added in the Methods section.

“Sample size determination

The sample size was calculated using the G*Power software for a priori analysis, with a sensitivity of 95% (�=0.05) and a study power of 90% (β=0.10). Since the patients were classified into three groups based on the dose, the effect size was determined by a one-way analysis of variance (ANOVA) using the effect size measure Cohen’s f. The resulting sample size n is 12. If 10% missing values were defined, the sample size (n=15 per group) is sufficient for the study.”

3. ". Assessment of clinical performance on the diagnosis and surgical planning by using 3D printing models created from low-dose CT images has not been reported in published studies."

Are you sure about this information? What are you referring to here? I can find many studies on related topics. Please, be more specific.

Response: We only found one paper that reported the use of 3D printed models created from low-dose CT images to evaluate the clinical performance of diagnosis and surgical planning. The sentence was changed as follows:

“Xiao et al. reported that 3D printed models created from low-dose CT images effectively evaluate the clinical performance of diagnosis and surgical planning. However, assessment of clinical performance on the diagnosis and surgical planning by using 3D printing models created from low-dose shoulder CT images has not been studied.”

Reference:

Xiao MQ , Zhang M , Lei M , et al. Application of ultra-low-dose CT in 3D printing of distal radial fractures. Eur J Radiol, 2020. 135: p. 109488. DOI：10.1016/j.ejrad.2020.109488

4. Was the 0.5 slice thickness, but and the other dimensions? please provide the resolution for each dimension of your voxel.

Response: In this study, we used 0.5 mm slice thickness (Table 1). 

5. "Patients’ 3D models of shoulder joints were reconstructed from the original CT data stored in DICOM format by using Vitrea fX"...

What is this? You need to use either a reference or named it with the name of the software, company, state, country etc.,

Response: The sentence was changed as follows: “Patients’ 3D models of shoulder joints were reconstructed from the original CT data stored in DICOM format by using the Vitrea fX software (Vitrea2，Canon, Japan).”

6. What was the resolution of the printed models in x,y and z? this information is essential to your paper.

Response: The following sentence was added:

“The resolution of the printed models was 0.011 mm � 0.011 mm � 0.0025 mm.”

7. "Two senior radiologists, with over 10 years of clinical experience and who were blinded to the results, performed quality assessment of CT images..."

What do you mean to say by senior? Were those radiologists part of a national board of radiologists? Faculty? Researchers? Clinicians? I need to know how to asses their expertise, rather than "senior"

Response: The following sentences were added.

“Two senior radiologists, who were part of a national board of radiologists and were clinicians with over 10 years of clinical experience for clinical diagnosis of musculoskeletal diseases performed quality assessment of CT images. They were blinded to the results of the study.”

8. "hey were blinded to the scanning parameters and rated the image quality based on a 3-point score system developed by published studies"

Please, specify which study using references.

Response: The sentence was changed as follows: “They were blinded to the scanning parameters and rated the image quality based on a 3-point score system as previously reported”

Reference

Xiao MQ , Zhang M , Lei M , et al. Application of ultra-low-dose CT in 3D printing of distal radial fractures. Eur J Radiol, 2020. 135: p. 109488. DOI: 10.1016/j.ejrad.2020.109488

9. "A 3-point rating system developed by published studies was employed for the assessment"

Same comment as before. If the study that you're referring to is the number 5, is this study (5) a validation study? It seems to be a validation study because you are using it as gold standard for your scores.

Response: The same paper (ref 5) was cited. 

The 3-point method is the scoring system used by the team before, and it is also widely used by other studies.

Reference

Xiao MQ , Zhang M , Lei M , et al. Application of ultra-low-dose CT in 3D printing of distal radial fractures. Eur J Radiol, 2020. 135: p. 109488. DOI: 10.1016/j.ejrad.2020.109488)

10. Statistical analysis:

Please, in addition to the ICC add a Bland-Altman test for inter-observer agreement for each group.

Response: Graded variables were used to test ICC. Classified data can not use Kendall Bland-Altman test.

11. "Three-dimension printing models of the shoulder joint created from ultra-low-dose CT scans can be used in the surgical planning of specific bones like the clavicle but perform insufficiently in the overall surgical planning for shoulder injuries due to the significant geometric flaws "

I don`t see that yous study has proven the hypothesis that it can be used for surgical planning, since you have not tested this. Please, re-write your conclusions in the paper and abstract based on your results only.

Response:

Three-dimension printing models of the shoulder joint created from ultra-low-dose CT scans can be used in the surgical planning of specific bones like the clavicle and nearly 94% of the clavicle models reached the adequate level. However, they performed insufficiently in the overall surgical planning for shoulder blade due to the significant geometric flaws with only 27% of the clavicle models reaching the adequate level.

12. "Gold standard of diagnosis performance: The gold standard of diagnosis performance is surgical findings or computed tomography/Diagnostic Radiology re-examination"

Please, provide more information, references, and how the gold standard was performed

Response: The following sentences were added.

“For fractures treated with surgery, the diagnosis was made by observing the fracture directly during surgery. For fractures by expectant treatment of fractures, the diagnosis was made based on the CT/DR review within 1-3 months: callus at the fracture end; dysplasia or old fracture without callus.”

13: "Radiation dose must increase to obtain high-definition CT images for developing accurate 3D printing models."

This is a general idea that should be state with caution, and not as an affirmation.

Response: The sentence was changed as follows:

“Radiation dose should increase in order to obtain high-definition CT images for developing accurate 3D printing models.”

14: Tables: The table names needs to contain the statistical test that you did. None of them has this information. "P < 0.05 means statistically significant" should go in the footnote.

Your ANOVA table contains no letters to distinguish differences between and among groups.

Response: Statistical significant differences were distinguished by letters (a, b, and c) in the tables

Reviewers' comments:

Reviewer's Responses to Questions

Comments to the Author

1. Is the manuscript technically sound, and do the data support the conclusions?

Reviewer #1: Yes

Reviewer #2: Yes

Reviewer #3: Yes

2. Has the statistical analysis been performed appropriately and rigorously?

Reviewer #1: Yes

Reviewer #2: Yes

Reviewer #3: Yes

3. Have the authors made all data underlying the findings in their manuscript fully available?

Reviewer #1: Yes

Reviewer #2: Yes

Reviewer #3: Yes

4. Is the manuscript presented in an intelligible fashion and written in standard English?

Reviewer #1: Yes

Reviewer #2: Yes

Reviewer #3: Yes

Response: Thanks for the positive comments.

5. Review Comments to the Author

Reviewer #1: Congratulations for the work, I believe that its relevance is justified as it seeks a way to reduce the radiation dose for patients and at the same time obtain a good diagnosis on a day-to-day basis.

- On the first paragraph of the introduction after the sentence “The dose of 50 radiation for traditional CT on a shoulder joint is extremely high with 5.28 mSv in 51 comparison to conventional radiography, which is less than 0.011 mSv...” I suggest adding the information of why you are studying the ultra-low dose CT and why it is so important to try to reduce the radiation dose in patients. Suggested references to justify the importance of studying the ultra-low dose protocol of CT:

Oestreich AE. RSNA centennial article: ALARA 1912: "As low a dose as possible" a century ago. Radiographics. 2014 Sep-Oct;34(5):1457-60. doi: 10.1148/rg.345130136. PMID: 25208291.

Mathews JD, Forsythe AV, Brady Z, Butler MW, Goergen SK, Byrnes GB, Giles GG, Wallace AB, Anderson PR, Guiver TA, McGale P, Cain TM, Dowty JG, Bickerstaffe AC, Darby SC. Cancer risk in 680,000 people exposed to computed tomography scans in childhood or adolescence: data linkage study of 11 million Australians. BMJ. 2013 May 21;346:f2360. doi: 10.1136/bmj.f2360. PMID: 23694687; PMCID: PMC3660619.

Response: The following sentences were added in the Introduction section.

“The risk of increased ionizing radiation exposure is the primary safety concern for having a CT scan. According to the data collected in the United States from 1991 to 1996, malignant tumors caused by CT radiation account for 0.4% of all malignant tumors.”

- The section “Patient selection”: Why did you include patients with osteoporosis in your study? As the bone density is altered in these individuals and their CT image can be different (or worse) from individuals without osteoporosis.

Response: The patients with osteoporosis were not included in the study. It is a typo, and was corrected as follows: “….non-osteoporosis (bone density: lumbar spine and/or hip joint T-score > -2.5).” 

- “Preparation of three-dimensional models”: In this section, information about the 3D printing method was missing, such as: type of 3D printing, reason for choosing this type of printing, diameter of the polylactic acid filament. The printing method was by extrusion (FDM) in your study, as I understand it, but we know that the printing quality using the Digital Lighting Process (DLP) method with vat polymerization is more accurate and faster (although it is more expensive). By adding this information, you can cite Eltes et al.1 study that compared 3D physical models printed with FDM or DLP, and demonstrated that the surface qualities, measured by roughness are adequate (~99% of values <0.1 mm) for both physical models, for their anatomic region. So, this justifies your choice for FDM 3D printing process. But, as in your article the printing was of low quality for some regions, it might be interesting for a next approach to try another type of 3D printing with greater precision (DLP, for example), for those regions that did not perform well with ultra-low dose CT 3D models.

o 1- Eltes, PE, Kiss, L, Bartos, M, et al. Geometrical accuracy evaluation of an affordable 3D printing technology for spine physical models. J Clin Neurosci 2020; 72: 438–446.

Response: The conventional orthopaedic surgery system does not have high requirements for 3D printing accuracy, which is enough to reach 0.1 mm, and extrusion 3D printing (FDM) is the cheapest of all 3D printing, which can meet the demand. 

- “Types of fractures in each group” section: Although the term comminuted fracture is generally understood, it would be interesting to briefly define the term after mentioning it for the first time.

Response: The sentence was changed as follows:

“In this study, shoulder fractures were divided into simple fractures and comminuted fractures (defined as fractures in which the bone was broken with 3 or more pieces) in terms of the strategy of treatment.”

- Lines 145 and 155: You mention the 3-point score system developed by “published studies”, but you only cite one study. If there are other studies, you must cite more then one (you cited only one – reference 5), and if you want to cite only this reference (5), then you should put this statement in the singular.

Response: The following references were cited.

Goetti R, Baumüller S, Feuchtner G, et al. High-pitch dual-source CT angiography of the thoracic and abdominal aorta: is simultaneous coronary artery assessment possible? [J]. American Journal of Roentgenology, 2010, 194(4): 938-944.

Peng A W, Dardari Z A, Blumenthal R S, et al. Very high coronary artery calcium (≥ 1000) and association with cardiovascular disease events, non–cardiovascular disease outcomes, and mortality: results from MESA [J]. Circulation, 2021, 143(16): 1571-1583.

- Discussion section:

o Line 223: You mention “Experimental results”, which are they? What is the reference for this affirmation?

o In your study, the ultra-low-dose protocol of the scapula and humerus on the 3D-printed model was inapplicable to clinical planning and failed its clinical application. Xiao et al.2, in 2021, obtained 3D printing models that, despite the quality being lower in the ultra-low dose group, were still sufficient to contribute to the preoperative evaluation and study, the diagnostic performance was not affected by the ultra-low dose protocol.

To what do you attribute this difference between studies? Do you attribute to the different anatomical regions being studied and their particularities? It would be interesting to add information like this to the discussion.

Response: The following sentences were added.

“We have previously demonstrated that the quality of CT images at the wrist joint meet the needs of clinical fracture diagnosis and the image quality of the 3D model created from ultra-low-dose CT scans is good enough for the surgical planning. However, in terms of the scapula part of the shoulder joint, the image quality of ultra-low dose CT meets the needs of clinical fracture diagnosis, but the image quality of the 3D model is too low poor for the surgical planning.”

According to our previous lumbar spine model, low-dose 3D printing model of the wrist joint, and this study, we speculate that it is related to the thickness of the cortical bone. Thicker cortical bone reduces image quality, and the boundary with the surrounding muscles is still clear. The cortical bone of the scapula body is thin, and the bone cortex of the clavicle wrist joint and lumbar vertebrae model is relatively thick. The image quality of the clavicle 3D printing model is better.

o Line 271:

FDM 3D printing, especially of large anatomical regions like the shoulder, take time, however, in terms of cost, it is currently much more affordable and 3D printing by extrusion (FDM) is the cheapest of all 3D printing. The advantage that 3D printing offers for this type of fracture, in terms of diagnosis and treatment accuracy, does it not outweigh the costs of 3D printing? I suggest you address this in the discussion. For example, Guochen Luo et al.3, for their anatomic region of interest, observed the following advantages from 3D printing technology assistance: less trauma, short operation time, less bleeding and reducing the difficulty of operation, which can reduce the waste of bone graft, and more complete reconstruction of the anatomical structure of the defective bone.

3- Luo G, Zhang Y, Wang X, Chen S, Li D, Yu M. Individualized 3D printing-assisted repair and reconstruction of neoplastic bone defects at irregular bone sites: exploration and practice in the treatment of scapular aneurysmal bone cysts. BMC Musculoskelet Disord. 2021 Nov 25;22(1):984. doi: 10.1186/s12891-021-04859-5. PMID: 34 

Response: The conventional orthopaedic surgery system does not have high requirements for 3D printing accuracy, which is enough to reach 0.1 mm. Extrusion 3D printing (FDM) is the cheapest of all 3D printing, and the 3D printing accuracy is 0.011 mm, which fully meets the demand. Higher precision 3D printing has too high cost and is not conducive to the research. The high cost made it difficult for 3D printing technology to be popularized in hospitals. The relatively low-cost material PLA (Tianwei Co., Ltd., US $9.83 for 1KG PLA material) was used in this study. Printing one 3D model costs about $2.16 and takes about 15.51 ± 1.20 hours. In our study, 3D printing process self-study videos were provided by 3D printing manufacturers, and 3D modeling parameters were adjusted remotely by the manufacturers. Ordinary computers could run the software, which was easy to learn.

Reviewer #2: This manuscript describes The clinical performance of ultra-low-dose shoulder CT scans: the assessment on image and physical 3D printing models.

I have some comments.

Material and Methods

patient selection

1.line 87. “osteoporosis (bone density: lumbar spine and/or hip joint T > -2.5).” . I think this is the definition of normal bone and osteopenia It could be write non-osteoporosis.

Response: Changed as suggested as follows: “non-osteoporosis (bone density: lumbar spine and/or hip joint T-score > -2.5)”.

2.line 88, 90 “T > -2.5” “T < -2.5”. It could be write as “T-score > -2.5, T-score < -2.5”

Response: Changed as suggested.

3. How about the pathological fracture patient? Were these patients included in the study?

Types of fractures in each group

Response: 

line 125. I think the definition of “avulsion fracture” means bone was avulsed by the ligament or tendon insertion. This description maybe confused by the reader. Please clarify this description.

Response: The sentence was changed as follows:

“Avulsion fractures refer to avulsion fractures caused by a sudden and violent contraction of a muscle or ligament, resulting in a long-diameter low than 5 mm wide bone piece.”

line 127. I think the definition of “dislocation” means the relationship of the joint. It could be write as “displacement of the bone cortex”.

Response: Changed as suggested.

Results

line 197. “(Table 2)” should be “(Table 3)”, please check it

Response: Changed as suggested.

Table 1

Age (year) “±” is redundant

Gender Dose the author mean male/female 15/15 10/20 13/20? Please clarify

Normal bone 14/16, 11/19/13/20, this description was confused by the reader, please clarify

Response: Table 1 was changed in the revised manuscript.

Reviewer #3: - The manuscript raises an interesting topic, but some corrections may make the information clearer.

- In the abstract it is described that the patients were randomly divided into two groups (standard-dose and ultra-low-dose groups), although in the materials and methods three are described. Please correct this information. 

Response: The sentence was changed as follows: “A total of 93 patients with displaced shoulder fractures were randomly divided into standard-dose, low-dose groups and ultra-low-dose groups.” 

- I suggest reviewing Table 01, there are different types of information. Please make it clearer.

Response: Table 1 was changed in the revised manuscript.

---

## [Decision Letter · Decision Letter 1]

18 Aug 2022

PONE-D-22-07553R1

The clinical performance of ultra-low-dose shoulder CT scans: the assessment on image and physical 3D printing models

PLOS ONE

Dear Dr. Xiao,

Thank you for submitting your manuscript to PLOS ONE. After careful consideration, we feel that it has merit but does not fully meet PLOS ONE’s publication criteria as it currently stands. Therefore, we invite you to submit a revised version of the manuscript that addresses the points raised during the review process.

We look forward to receiving your revised manuscript.

Kind regards,

Jonas Bianchi, DDD, MS, Ph.D

Academic Editor

PLOS ONE

Journal Requirements:

Reviewers' comments:

Reviewer's Responses to Questions

**Comments to the Author**

1. If the authors have adequately addressed your comments raised in a previous round of review and you feel that this manuscript is now acceptable for publication, you may indicate that here to bypass the “Comments to the Author” section, enter your conflict of interest statement in the “Confidential to Editor” section, and submit your "Accept" recommendation.

Reviewer #1: All comments have been addressed

Reviewer #2: (No Response)

2. Is the manuscript technically sound, and do the data support the conclusions?

Reviewer #1: Yes

Reviewer #2: Yes

3. Has the statistical analysis been performed appropriately and rigorously? 

Reviewer #1: Yes

Reviewer #2: Yes

4. Have the authors made all data underlying the findings in their manuscript fully available?

Reviewer #1: Yes

Reviewer #2: Yes

5. Is the manuscript presented in an intelligible fashion and written in standard English?

Reviewer #1: Yes

Reviewer #2: Yes

6. Review Comments to the Author

Reviewer #1: (No Response)

Reviewer #2: This manuscript describes the clinical performance of ultra-low-dose shoulder CT scans: the assessment on image and physical 3D printing models.

I have two comments.

(1). Material and Methods: patient selection

How about the pathological fracture patient? Were these patients included in the study? The authors did not answer this question.

(2). line 143-147 : ” Displaced fractures refer to avulsion fractures caused by a sudden and violent contraction of a muscle or ligament, resulting in a long-diameter low than 5 mm wide bone piece. Non-displaced fractures are defined as no angulation or shortening, less than 2 mm wide fracture line and/or less than 1 mm displacement of the bone cortex [21].”

This description will still confuse the reader. Please clarify the definition. Displaced fracture is not equal to the avulsion fracture. How much displacement (distance) or angulation (angle) was defined as displaced fracture in this study? Please clarify the definition.

7. PLOS authors have the option to publish the peer review history of their article (what does this mean?). If published, this will include your full peer review and any attached files.

Reviewer #1: No

Reviewer #2: No

---

## [Author Response · Author response to Decision Letter 1]

26 Aug 2022

Dear Dr. Bianchi,

Thank you very much for your decision letter and advice on our manuscript (Manuscript No. PONE-D-22-07553) entitled “The clinical performance of ultra-low-dose shoulder CT scans: the assessment on image and physical 3D printing models”. We also thank the reviewers for the constructive comments and suggestions. We have revised the manuscript accordingly, and all amendments are indicated by red font in the revised manuscript. In addition, our point-by-point responses to the comments are listed below this letter.

This revised manuscript has been edited and proofread by Medjaden Inc..

We hope that our revised manuscript is now acceptable for publication in your journal and look forward to hearing from you soon. 

With best wishes,

Yours sincerely,

Mengqiang Xiao

First of all, we would like to express our sincere gratitude to the reviewers for their constructive and positive comments.

Reviewer #2: This manuscript describes the clinical performance of ultra-low-dose shoulder CT scans: the assessment on image and physical 3D printing models.

I have two comments.

(1). Material and Methods: patient selection

How about the pathological fracture patient? Were these patients included in the study? The authors did not answer this question.

Response: Patients with pathological fractures were not included in the study. The following sentences were added to the Materials and Methods section.

“Exclusion criteria were as follows: less than 18 years of age, pregnancy, patients who refused to participate in the study, and patients with osteoporosis (bone density: lumbar spine and/or hip joint T-score < -2.5) or pathological fractures.”

(2). line 143-147:” Displaced fractures refer to avulsion fractures caused by a sudden and violent contraction of a muscle or ligament, resulting in a long-diameter low than 5 mm wide bone piece. Non-displaced fractures are defined as no angulation or shortening, less than 2 mm wide fracture line and/or less than 1 mm displacement of the bone cortex [21].”

This description will still confuse the reader. Please clarify the definition. Displaced fracture is not equal to the avulsion fracture. How much displacement (distance) or angulation (angle) was defined as displaced fracture in this study? Please clarify the definition.

Response: The following sentences were added.

Line 143-147: “Non-displaced fractures are defined as having no angulation or shortening, a fracture line less than 2 mm wide and/or less than 1 mm displacement of the bone cortex. Displaced fractures are defined as having a fracture line more than 2 mm wide and/or more than 1 mm displacement of the bone cortex. Avulsion fractures caused by a sudden and violent contraction of a muscle or ligament, were grouped into non-displaced fractures or displaced fractures when the fracture had a long-diameter � 5 mm or > 5 mm wide bone piece, respectively [21].”

---

## [Decision Letter · Decision Letter 2]

13 Sep 2022

The clinical performance of ultra-low-dose shoulder CT scans: the assessment on image and physical 3D printing models

PONE-D-22-07553R2

Dear Dr. Xiao,

We’re pleased to inform you that your manuscript has been judged scientifically suitable for publication and will be formally accepted for publication once it meets all outstanding technical requirements.

Kind regards,

Jonas Bianchi, DDD, MS, Ph.D

Academic Editor

PLOS ONE

Reviewer's Responses to Questions

**Comments to the Author**

1. If the authors have adequately addressed your comments raised in a previous round of review and you feel that this manuscript is now acceptable for publication, you may indicate that here to bypass the “Comments to the Author” section, enter your conflict of interest statement in the “Confidential to Editor” section, and submit your "Accept" recommendation.

Reviewer #2: All comments have been addressed

---

## [Editor Report · Acceptance letter]

16 Sep 2022

PONE-D-22-07553R2 

The clinical performance of ultra-low-dose shoulder CT scans: the assessment on image and physical 3D printing models 

Dear Dr. Xiao:

I'm pleased to inform you that your manuscript has been deemed suitable for publication in PLOS ONE. Congratulations! Your manuscript is now with our production department. 

Kind regards, 

on behalf of

Dr. Jonas Bianchi 

Academic Editor

PLOS ONE